# Immobilization of Dextranase on Nano-Hydroxyapatite as a Recyclable Catalyst

**DOI:** 10.3390/ma14010130

**Published:** 2020-12-30

**Authors:** Yanshuai Ding, Hao Zhang, Xuelian Wang, Hangtian Zu, Cang Wang, Dongxue Dong, Mingsheng Lyu, Shujun Wang

**Affiliations:** 1Jiangsu Key Laboratory of Marine Bioresources and Environment, Jiangsu Ocean University, Lianyungang 222005, China; dingys@jou.edu.cn (Y.D.); zhanghao0505@jou.edu.cn (H.Z.); xlwang@jou.edu.cn (X.W.); htzu@jou.edu.cn (H.Z.); wangcang@jou.edu.cn (C.W.); dxdong@jou.edu.cn (D.D.); mslyu@jou.edu.cn (M.L.); 2Co-Innovation Center of Jiangsu Marine Bio-industry Technology, Jiangsu Ocean University, Lianyungang 222005, China; 3Collaborative Innovation Center of Modern Biological Manufacturing, Anhui University, Hefei 230039, China

**Keywords:** dextranase, hydroxyapatite, immobilization, recyclable, plaque removal

## Abstract

The immobilization technology provides a potential pathway for enzyme recycling. Here, we evaluated the potential of using dextranase immobilized onto hydroxyapatite nanoparticles as a promising inorganic material. The optimal immobilization temperature, reaction time, and pH were determined to be 25 °C, 120 min, and pH 5, respectively. Dextranase could be loaded at 359.7 U/g. The immobilized dextranase was characterized by field emission gun-scanning electron microscope (FEG-SEM), X-ray diffraction (XRD), and Fourier-transformed infrared spectroscopy (FT-IR). The hydrolysis capacity of the immobilized enzyme was maintained at 71% at the 30th time of use. According to the constant temperature acceleration experiment, it was estimated that the immobilized dextranase could be stored for 99 days at 20 °C, indicating that the immobilized enzyme had good storage properties. Sodium chloride and sodium acetic did not desorb the immobilized dextranase. In contrast, dextranase was desorbed by sodium fluoride and sodium citrate. The hydrolysates were 79% oligosaccharides. The immobilized dextranase could significantly and thoroughly remove the dental plaque biofilm. Thus, immobilized dextranase has broad potential application in diverse fields in the future.

## 1. Introduction

Dextranase (1,6-α-D-glucan 6-glucanohydrolase; EC 3.2.1.11) specifically hydrolyzes α-1,6 glycosidic bond of dextran with broad application prospects in diverse industries [1,2,3], such as in the production of low molecular-weight medical dextran and in food processing, sugarcane factories, and the prevention and treatment of dental plaque biofilms [4,5,6].

The development of dextranase-immobilization technology is expected to provide a promising technology for its reuse [7,8]. The purpose of immobilization is to immobilize an enzyme inside or on the surface of an insoluble material such that the catalytic activity of the enzyme is maintained. The immobilization strategies generally focus on continuous operation, increasing stability, and reducing the usage of enzymes. The feasibility of immobilizing dextranase on different support materials has been verified via physical adsorption, ion adsorption, encapsulation, and covalent adsorption. However, it remains a challenge to identify materials that provide the requirements of special characteristics for higher enzyme immobilization efficiency. Moreover, the robustness and costs of immobilization are significant factors that affect their application. Ideally, the carrier material should have a relatively high specific surface area to enable immobilization of a large amount of enzyme [9,10]. Moreover, hydrophilicity with good infusibility and low solubility of the substrate could avoid product contamination. From the immobilization of biocatalysts to the recovery of enzymes, the mechanical resistance and thermal stability of the scaffolds are important throughout the process [11]. Another important parameter to be evaluated during the fixation process is the diffusion rate of the substrate to the catalyst. Hence, the identification of suitable materials for different enzymes remains the focus of attention of the researchers [12,13].

Hydroxyapatite (HA), a non-toxic nano-material, is a promising inorganic material that immobilizes enzymes [14,15]. Adsorption of proteins on HA has been applied in a variety of applications such as in the chromatographic separation of proteins, bone regeneration in biomedical science, and drug delivery in the pharmaceutical industry [16,17]. HA structures is composed of phosphate and calcium groups that are involved in ion adsorption of protein-side groups. The presence of calcium ions in HA can also be complexed with formic acid. The chemical group present in the amino acid of the enzyme contributes to the high stability and fixed interaction characteristics. Recent studies have reported the immobilization of lipase, β-glucosidase, and dextransucrase on HA nanoparticles [18,19,20]. However, their research is not comprehensive enough. This paper systematically studied the conditions, properties, utilization, and storage stability of HA-immobilized dextranase. Considering the large-scale industrial application of biocatalysts, HA nanoparticles need to be further explored as immobilized enzyme carrier materials [21,22].

Dental plaque is a biofilm structure composed of a variety of microorganisms that can lead to the development of caries and periodontal diseases. *S. mutans* is considered as one of the important bacteria that form dental plaque and dental caries [23]. The enzymatic hydrolysis of dental plaque has become a hot research topic in the recent years owing to the mild nature of the process and the high efficiency. In this study, dextranase was immobilized on HA under different conditions and the optimal conditions of immobilization yield was investigated. Moreover, the characteristics of HA-immobilized dextranase were investigated, and the effect of dextranase on plaque removal was discussed. We obtained an effective method to ensure that immobilized dextranase has high catalytic activity, stability, and recyclability.

## 2. Materials and Methods

### 2.1. Materials

*Arthrobacter oxidase* KQ11 was used to prepare dextranase in our laboratory (Strain KQ11 was inoculated in the fermentation medium. After 28 h of fermentation, the broth was centrifuged at 12,000× *g* at 4 °C for 10 min. The supernatant contained the crude enzyme, which was then ultrafiltered through a 30-kDa membrane. Then, the dextranase was lyophilized to obtain the enzyme powder). Nano-HA (<100-nm particle size) was purchased from Shanghai Aladdin Biochemical Technology Co., Ltd. (Shanghai, China). Phenol from Nanjing Chemical Reagent Co., LTD. (Nanjing, China). Crystal violet from Tianjin Fuchen Chemical Reagent Factory (Tianjin, China). NaCl, NaF, sodium acetate, sodium citrate, acetic acid, sodium hydroxide, 3,5-dinitrosalicylic acid, dextran 20,000, and hydrochloric acid were bought from Sinopharm Chemical Reagent Co., Ltd. (Shanghai, China). All other reagents used in the study were of analytical grade.

### 2.2. Immobilization of Dextranase on HA

HA and dextranase were mixed in tubes, and the fixation process was monitored by measuring the enzyme activity in the supernatant obtained by centrifugation at 13,800× *g* for 5 min. Free dextranase served as the control. The relative activity was calculated based on the average measurement [22].

The percent immobilized yield (IY) was calculated by using the following formula:Immobilization yield (%) = (1 − S/C) × 100
where, S and C represent the activities of supernatant and control, respectively.

For dextranase activity determination, 50 μL of the enzyme solution and 150 μL of 3% dextran (molecular weight: 20,000; 50 mM; pH 6 sodium acetate buffer preparation) were mixed and placed on a 60 °C water bath for 15 min. Then, 200 μL of 3,5-dinitrosalicylic acid preparation reagent (DNS) was added to stop the reaction, followed by boiling for 5 min and the addition of 3 mL of distilled water. The absorbance of the mixture was then read at 540 nm using a microplate reader (Infinite M1000 Pro; Tecan, Männedorf, Switzerland). For the immobilized enzymes, 10 mg of HA-immobilized dextranase was added to 1 mL of the substrate solution and subjected to reaction under the same conditions, after which the enzyme activity was estimated.

One unit (U) of enzyme activity was defined as the amount of enzyme required to release 1 μmol of maltose in a minute under the condition of 60 °C and pH 6, using 3% dextran 20,000 as the substrate.

### 2.3. Optimization of Immobilizing Parameters

In order to determine the ideal conditions for the fixation process, the optimal reaction temperature was determined. For this purpose, dextranase powder was dissolved in pure water, and the resultant enzyme solution was used for immobilization reaction under different temperature conditions (10–45 °C) for 1 h. The reaction was performed for different time periods (5–240 min) at 25 °C. In addition, different concentrations of dextranase (0.5, 1.0, 1.5, 2.0, and 2.5 mg/mL) were used for the immobilization purpose. Similarly, different pH conditions and buffers were selected in the optimization study (sodium acetate buffer at pHs 4–6 and Tris-HCl buffer at pHs 7–9).

### 2.4. The Properties of Immobilized Dextranase

The immobilized dextranase was prepared under optimal conditions, and their properties were evaluated. The temperature was set to 20–65 °C, while the pH was set between 4 (acetic sodium buffer for 4–6) and 9 (Tris-HCl buffer for 7–9). The relative activity was calculated, and all experiments were performed in triplicate.

### 2.5. Desorption by Sodium Salts

The desorption of HA-immobilized dextranase was evaluated with different eluents. Different sodium salts, such as sodium chloride, sodium acetate, sodium fluoride, and sodium citrate were used, and the elution effect of different concentrations of selected sodium salt on the immobilized dextranase was investigated. The volume of the buffer used for elution was the same as that used for immobilization.
Elution efficiency (%) = E/A × 100
where, E = dextranase activity of eluent and A = the activity of immobilized dextranase

### 2.6. Characterization Methods

Field emission gun-scanning electron microscope (FEG-SEM) was used to analyze the morphological particles of HA. Briefly, the dextranase was mixed with HA for 60 min. The resultant solution was centrifuged and washed thrice with pure water, and the precipitate was resuspended in pure water on a shaker. Finally, the sample was lyophilized by drying directly at 60 °C for 12 h). The crystallinity of HA particles was measured using X-ray diffraction (XRD). The diffraction pattern was recorded in the diffraction range (2θ) of 20°–70°. Fourier-transformed infrared spectroscopy (FT-IR) was utilized to analyze HA particles without or with dextranase. The measurement was performed in the transmission mode in the mid-infrared range of 400–4000 cm^−1^.

### 2.7. The Hydrolysates of Immobilized Dextranase

The substrate (1%) was mixed with immobilized dextranase and allowed to react for 2 h. Then, the solution was centrifuged at 13,800× *g* for 2 min, and the hydrolysates were obtained in the supernatant and subjected to analyses by high performance liquid chromatography (HPLC).

### 2.8. Stability of Reuse and Storage

The reusability of the immobilized dextranase was evaluated by assessing the activity of immobilized dextranase under optimal conditions. Next, the immobilized enzyme was obtained by centrifugation (13,800× *g*, 2 min). Then, the precipitate was washed with pure water and resuspended in fresh substrate for the next step. For the stability test, a constant temperature-accelerated experiment was performed to estimate the storage period of the immobilized dextranase. The immobilized dextranase was incubated at 25, 35, and 45 °C for 6 days, and the activity was monitored every day. The first-level pyrolysis diagram of the immobilized dextranase was obtained from plotting the enzyme activity logarithm and the incubation time. Then, the thermal degradation K was obtained from the equivalent of the linear correlation and regression analysis. The K was plotted against the number of days required for enzyme inactivation at different temperatures. Finally, the storage life of the immobilized dextranase was estimated.

### 2.9. Removal of the Dental Plaque Biofilm

The effect of dextranase on the degradation of biofilm produced by *Streptococcus mutans* (ATCC 25175) was evaluated. For this purpose, *S. mutans* was first cultured in sucrose-free brain heart infusion (BHI) broth medium at 37 °C for 18 h. Then, 20 μL of the broth was inoculated into 180 μL of the BHI medium supplemented with 1% sucrose in a 96-well plate, followed by incubation at 37 °C for 24 h under anaerobic condition. Next, the broth was gently aspirated, and the wells were washed thrice with 300 μL of 0.2 mM phosphate buffer (PB). After the wells were allowed to dry naturally in the air for 1 h, dextranase, HA, and immobilized dextranase were added to the wells. The amount of free dextranase was the same as the amount of immobilized dextranase; similarly, the amount of HA was the same as the amount of HA in immobilized dextranase. Pure water served as the control. The wells were washed thrice after the 1-h incubation. After natural drying for 1 h, 200 μL of 0.1% crystal violet was added to the well. After 5 min, the dye was removed, and the wells were thrice washed by PB. After natural drying, 200 μL of 95% ethanol was added to re-dissolve the stained biofilm, and the solution was subjected to absorbance measurement at 595 nm. The rate of biofilm degradation was then calculated.

To prepare the sample for SEM observation, sterile glass slides were arranged in a 24-well plate, and *S. mutans* was cultured in the wells. At the end of the experiment, the slides were removed and fixed in 2.5% glutaraldehyde at 4 °C for overnight. Next, the slides were subjected to different concentrations of ethanol gradient dehydration (from 50% to 100%) for 5 min. Finally, the slides were dried, sprayed with gold, and observed under SEM (model JFC-1600; JSM-6390LA; JEOL, Tokyo, Japan).

## 3. Results

### 3.1. Immobilization Condition

Our results revealed that the immobilization yield remained stable at 25–35 °C (Figure 1a). In addition, lower temperature of 10–20 °C was not found to be conducive for immobilization. The reaction time was at least 2 h (Figure 1b), and 81% dextranase was immobilized on HA. The immobilization yield was the highest at 1 mg/mL dextranase powder concentration (Figure 1c). Based on these results, 1 mg HA was found to immobilize 100 μg of dextranase powder. Although higher concentration of dextranase could increase the immobilization, the immobilization yield was found to decrease (Figure 1c). The immobilization rate could reach 90% at pH 5, and the immobilization condition was preferred under acidic buffer than under alkaline buffer (Figure 1d). HA nanoparticles demonstrated a negative potential in the pH range of 4–9 (Appendix A). In the process of immobilization, we noted that the fixation efficiency decreased with higher pH value. The highest immobilized yield was performed at pH 5, at the potential HA of −7 mV. The isoelectric point of dextranase was 5.19 [24], and the dextranase had a weak negative charge at pH 5. Accordingly, the acidic conditions were found to enhance the immobilization yield of the dextranase.

Two different buffers were used for immobilization to optimize the performance. Notably, the relative dextranase activity was the highest at pH 5. In fact, the optimal pH of KQ11 dextranase was 7. Our experiments demonstrated an efficient process for dextranase immobilization on HA in a short time.

### 3.2. Characteristics of Immobilized Dextranase

The immobilization was influenced by the properties of dextranase. For instance, the immobilized dextranase was more stable on a wide range of pH (Figure 2a). The optimal pH shifted from pH 6 to 5 (Figure 2b). The temperature condition remained the same (Figure 2c). However, a significant improvement was noted at 65 °C, at which the immobilized dextranase maintained 90% of its activity. On the contrary, dextranase could not be measured at 65 °C (Figure 2d). Based on our results, HA may enhance the ability of enzymes against higher temperature for certain time periods.

### 3.3. Desorption by Sodium Salts

Immobilized dextranase could not be desorbed in a solution of 0.4 M sodium chloride and sodium acetic (Figure 3a,c) [18]. These observations indicate a non-universality of ionic interactions that can be conducted at high concentrations in the presence of these salts. The strong physical adsorption of HA with dextranase has important industrial significance considering that the enzyme may not be easily leached from the carrier under the reaction conditions [25,26]. In contrast, dextranase could be desorbed in a solution of 50 mM sodium fluoride and sodium citrate (Figure 3b,d). Considering that citrate is composed of three carboxylic acid groups and only one sodium acetate group; this finding provides strong evidence that the COO-site of the enzyme was critical for their interaction with HA nanoparticles. The effect of these carboxylic acids can be attributed to the effect of the polar groups on the Ca^2+^ ions of the HA nanoparticles, resulting in a certain type of coordination reaction. Calcium ions acted as stable coordinators of the hard acid with an oxygen atom of a hard base.

Hydrogen bonds, ionic bonds, coordinate, and van der Waals force may have important contribution to the process of immobilization. The structure of HA is composed of phosphates, hydroxyls, and calcium groups, which are useful for ionic interactions with proteins. Our results thus suggested that the main acting force was ionic bonds [10,27].

Proteins can be adsorbed via a coordination bond established with the remaining position of the chelating metal ion on the surface of the substrate, such as in immobilized metal ion affinity chromatography (IMac). Calcium ions are not sequestered by complexing agents because the metal is already a part of the crystal structure of the HA. The high stability of the complex formed may be attributed to the ring structure formation, similar to that in a chelation reaction. Other factors that may be involved include electrostatic forces, van der Waals forces, and hydrogen bonding of the OH-groups of HA, although it remains difficult to determine their relative contributions [9,28,29].

### 3.4. Detection of Immobilized Dextranase

The surface of HA showed smooth after immobilization of dextranase, and it tended to agglomerate, indicating that they were bound together by relatively weak forces (Figure 4a,b). We noted that the immobilized dextranase was easy to disperse. On the other hand, the HA surface possessed a certain roughness, which made it difficult to estimate the thickness of dextranase adsorbed onto the surface.

Figure 4c displays the results of FT-IR. The FT-IR spectrum showed a stronger band at 1648 cm^−1^, which corresponded to a symmetric C=O stretching absorption. This observation suggested the presence of a carboxylic acid group of the amino acid. As the enzyme adsorbed, the band corresponding to the C=O bond of the amino acid (1650 cm^−1^) became slightly enhanced. The peaks located at 1421 and 1648 cm^−1^ were attributed to N-H and C=O stretching vibrations, respectively. These results signify the successful immobilization of dextranase on HA nanoparticles. Agrawal et al. also confirmed the immobilization of dextranase (SiO_2_ nanoparticles) by FT-IR, reporting 1648 cm^−1^ absorbance of the amino acid group of the enzyme [30]. Thus, the absorption bands of hydroxyl and phosphate groups were demonstrated in the HA molecule (Ca_10_(PO_4_)_6_(OH)_2_). Similarly, the bands 1421 and 1648 cm^−1^ corresponded to the presence of C=O in the HA particles.

The XRD results suggested that the crystal form of HA did not change after its immobilization. The peaks of hydroxyapatite with dextranase were strengthened, and the relative intensity of each peak had a good corresponding relationship with the XRD pattern of hydroxyapatite (Figure 4d).

### 3.5. The Hydrolysates and Recyclable Catalysts

The hydrolysate was analyzed by HPLC. The immobilized dextranase maintained excellent activity by hydrolyzing dextran to produce oligosaccharides. The concentration of maltotriose, maltopentaose, and maltohexaose were 79.29% (Figure 5a).

Our results revealed that the immobilized dextranase could maintain 84.02% of its hydrolysis capacity at the 20th time and 71.42% at the 30th time of use (Figure 5b). Thus, the immobilized dextranase expressed remarkable repeatability. Moreover, according to the constant temperature acceleration experiment, there is a linear relationship between immobilized enzyme activity and storage time at different incubated time (Figure 5c). It was estimated that the immobilized dextranase could be stored at 20 °C for 99 days (Figure 5d). The stability of the immobilized dextranase was thus significant.

### 3.6. Degradation of Dental Plaque Biofilm

Crystal violet staining in the present experiment revealed that immobilized dextranase was the best for removing dental plaques. However, the immobilized dextranase could achieve 86.44% degradation of the biofilm, showing an obvious significant effect on the *S. mutans* biofilm (Table 1).

The SEM results confirmed that the immobilized dextranase significantly influenced the removal of dental plaque formed by *S. mutans.* As shown in Figure 6, the slide is covered by *S. mutans* fully in the control. Dextranase could definitely remove the plaque, and the extent of removal depended on the quantity of dextranase. However, the biofilm still contained excessive residues after the addition of HA. Remarkably, the immobilized dextranase could avoid the adhesion of *S. mutans* on the slide (Figure 6d). In fact, the biofilm was thoroughly removed with the use of immobilized dextranase, which demonstrated stable property and showed a potential for application in dental care [31,32].

## 4. Conclusions

We conducted a systematic study to evaluate the feasibility of immobilizing dextranase on HA nanoparticles. The optimal immobilization conditions were inferred as follows: Temperature, pH, and reaction time of 25 °C, 5, and 120 min, respectively. Dextranase could be loaded at the rate of 359.73 U/g of HA. The property of the immobilized dextranase induced a slight change in the optimal pH and thermostability. The immobilized dextranase was characterized by FEG-SEM, XRD, and FT-IR. Our results demonstrated that dextranase could be immobilized on the surface of nano-HA. Sodium chloride and sodium acetic could not desorb the immobilized dextranase. In contrast, dextranase could be desorbed by sodium fluoride and sodium citrate. After 30-times of the hydrolysis cycles, the immobilized dextranase retained approximately 71.42% of its initial activity as well as it was estimated that the immobilized dextranase could be stored for 99 days at 20 °C and for 93 days at 30 °C. The hydrolysates were 79.29% oligosaccharides, and the immobilized dextranase could significantly and thoroughly remove the dental plaque biofilm. The successful immobilization of dextranase and its performance indicates its exciting and promising application prospects in the future.

## Figures and Tables

**Figure 1 materials-14-00130-f001:**
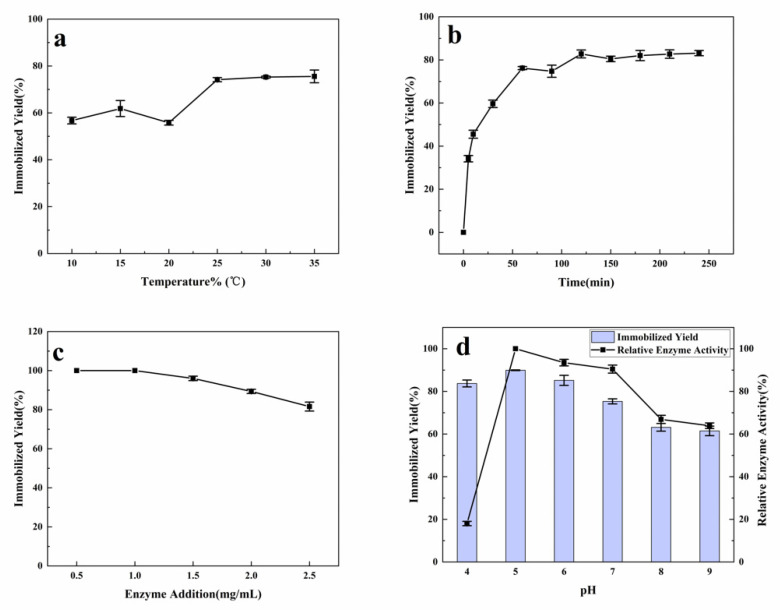
The effect of enzyme immobilization process under different immobilization conditions of (**a**) temperature, the reaction took 1 h (**b**) time, the reaction temperature was 25 °C, (**c**) enzyme concentration, and (**d**) pH.

**Figure 2 materials-14-00130-f002:**
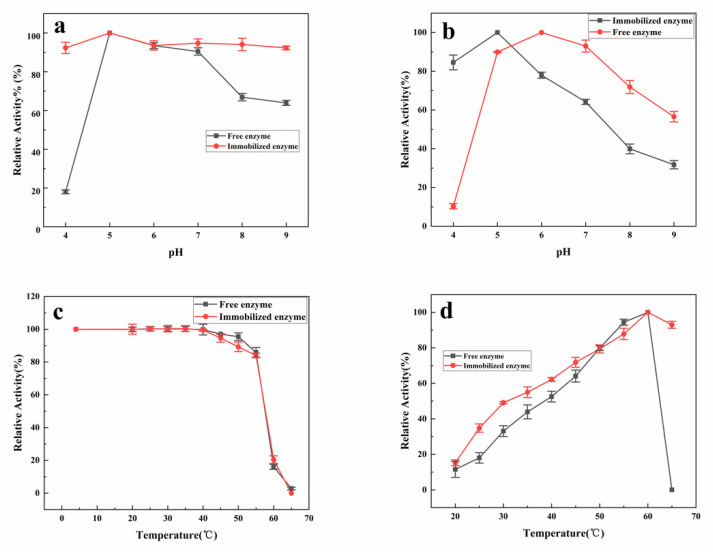
Properties of immobilized dextranase for optimal activity: (**a**) pH stability, (**b**) optimal pH of activity, (**c**) temperature stability, and (**d**) optimal temperature of activity.

**Figure 3 materials-14-00130-f003:**
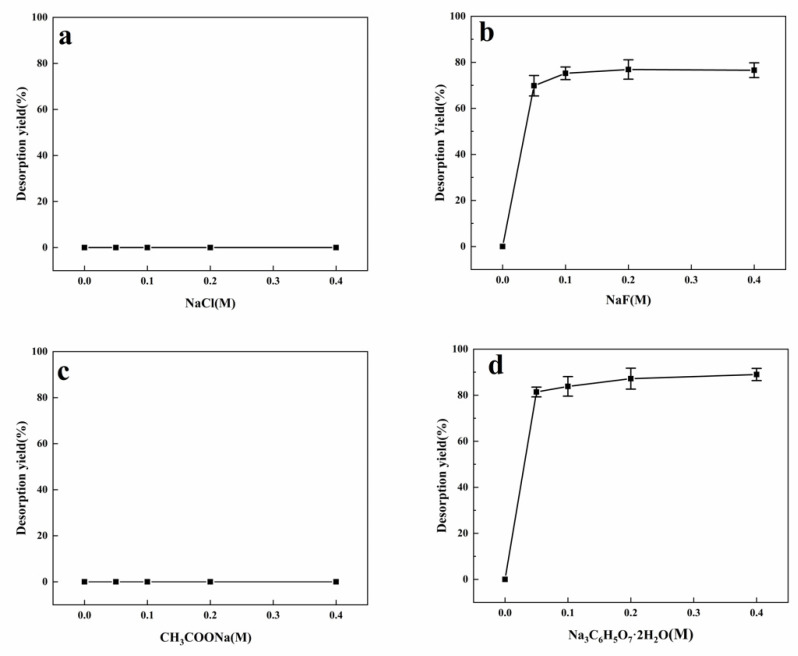
Desorption of dextranase from hydroxyapatite (HA) by (**a**) NaCl, (**b**) NaF, (**c**) CH_3_COONa, and (**d**) Na_3_C_6_H_5_O_7_.

**Figure 4 materials-14-00130-f004:**
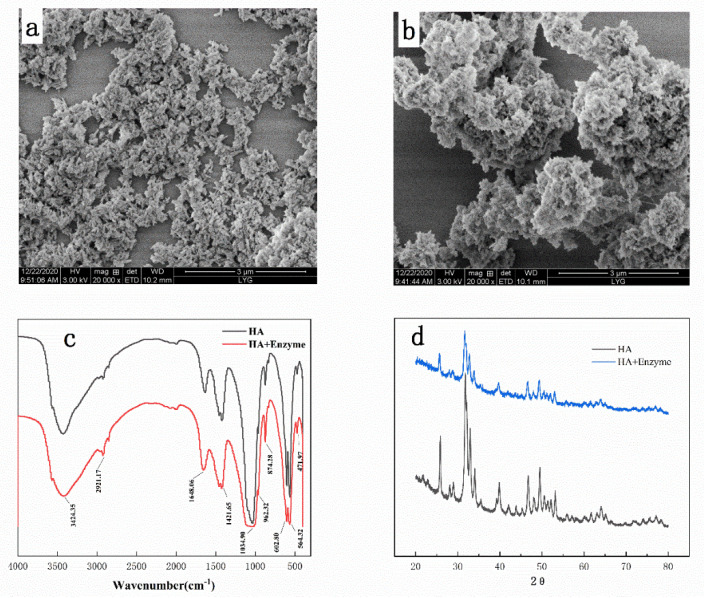
Detection of HA and immobilized dextranase; (**a**) field emission gun-scanning electron microscope (FEG-SEM) images of HA, (**b**) FEG-SEM images of immobilized dextranase, (**c**) FT-IR spectra, and (**d**) XRD spectra.

**Figure 5 materials-14-00130-f005:**
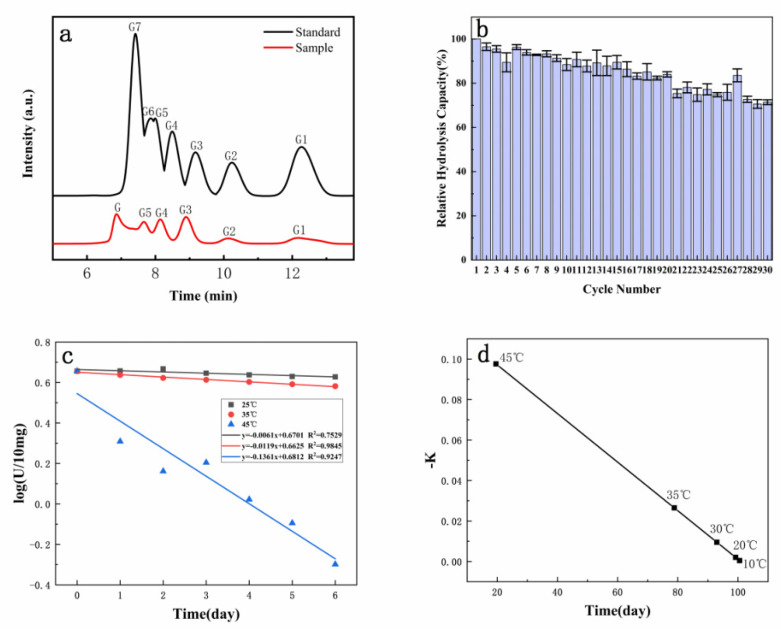
(**a**) HPLC of standard G1-G7: Glucose, maltose, maltotriose, maltotetraose, maltopentaose, maltohexaose, maltoheptaose, and HPLC analysis of hydrolysates; (**b**) reusing experiment of immobilized dextranase, (**c**) simulated first-level pyrolysis diagram of immobilized enzyme, and (**d**) pseudo first-order pyrolysis rate of immobilized enzyme and the time required for inactivation.

**Figure 6 materials-14-00130-f006:**
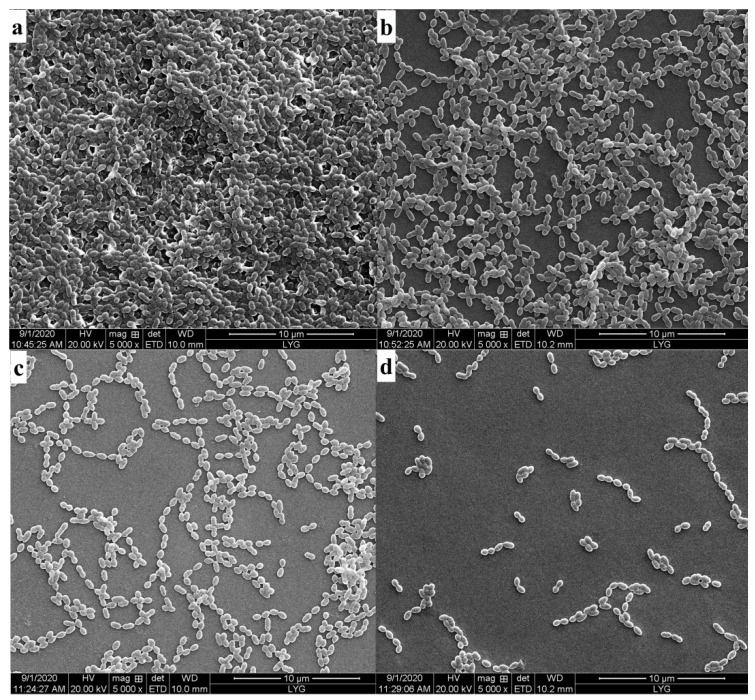
(**a**) Control group (add pure water); (**b**) add free enzyme (3.6 U); (**c**) add HA (10 mg); and (**d**) add immobilized enzyme (10 mg HA immobilized 3.6 U of free enzyme).

**Table 1 materials-14-00130-t001:** The removal rate of different additives on the dental plaque biofilm.

Additives	Biofilm Removal Rate
Control	0.00 ± 8.66
Free Enzyme	14.32 ± 12.29
HA	52.56 ± 9.64
Immobilized Enzyme	86.44 ± 1.01

## Data Availability

The interaction data between HA and dextranase used to support the findings of this study are included within the article and the Appendix A. Also, all the data used to support the findings of this study are available from the corresponding author upon request.

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
