# Peer review of "Immobilization of Dextranase on Nano-Hydroxyapatite as a Recyclable Catalyst"

_materials, 2020, doi:10.3390/ma14010130_

Round 1

Reviewer 1 Report

The work deals with the dextranase immobilization onto hydroxyapatite nanoparticles in different temperatures, reaction times and pH to optimize the immobilization process. After reaching the higher enzyme immobilization efficiency, it was tested against S. mutans to relieve biofilm removal for the potential dental plague removing application. The paper is well structured, and it can be interesting for the readers, however, some corrections are needed, before publishing, as follows:

C1) In section 2.2 and 2.9, the abbreviations of DNS and BHI needs to be explained before using it.

C2) In the Results section, the heading of “3.1” is missing. As authors used sub-sections of 3.2, 3.3, etc, would be better to add a heading for 3.1. 

C3) In section 3.2, “The temperature condition remained the same (Figure 2b).” does not refers to Figure 2b, it should be changed to Figure 2c.

C4) In section 3.3, “Immobilized dextranase could not be desorbed in a solution of 0.4 M sodium chloride and sodium acetic[23, 24].” The sentence does not refer to references of 23 and 24. Please find the related part in the mentioned reference papers, or change the references with relevant ones.  

C5) In figure 4, the scale of the SEM images are different. In the general view, it seems that dextranase is fully immobilized, while HA is partially immobilized, however, its magnification is 100 times smaller. Please choose the same magnification and change it.

C6) In section 3.6, the authors mentioned that “Crystal violet staining in the present experiment revealed that immobilized dextranase was the best for removing dental plaques (Figure 6). Interestingly, HA was found to be more effective in dental plaque removal than dextranase.”. According to Table 1 and Figure 7, the dextranase immobilized sample was more effective among all, which is confirmed by the first sentence above. However, the second sentence refers to HA was found the most effective. Please correct this mismatch.

C7) Table 1, Figure 3, Figure 5a and Figure 5b were not mentioned in the text. Please add relevant sentences to refer to them.

C8) The conclusion part is missing in the manuscript. Please add it.

Author Response

Dear Editor and Reviewers,

We have read these comments carefully, and we are really appreciated for you kindly comments. We have revised our manuscript according to the suggestions. The amend parts have been marked as yellow background in the manuscript. The main corrections in the paper and the responds to the reviewer’s comments are as following:

(1) In section 2.2 and 2.9, the abbreviations of DNS and BHI needs to be explained before using it.

Answer:thanks for your comments. According to your suggestion, we have added relevant content in the manuscript. ‘ 3,5-Dinitrosalicylic acid preparation reagent(DNS)Brain Heart Infusion Broth (BHI)’

(2) In the Results section, the heading of “3.1” is missing. As authors used sub-sections of 3.2, 3.3, etc, would be better to add a heading for 3.1. 

Answer:thanks for your comments. According to your suggestion, we have added relevant content in the manuscript.: ‘3.1 Immobilization condition’.

(3) In section 3.2, “The temperature condition remained the same (Figure 2b).” does not refers to Figure 2b, it should be changed to Figure 2c.

Answer:thanks for your comments. We have done as your comments in the manuscript.

94) In section 3.3, “Immobilized dextranase could not be desorbed in a solution of 0.4 M sodium chloride and sodium acetic [23, 24].” The sentence does not refer to references of 23 and 24. Please find the related part in the mentioned reference papers, or change the references with relevant ones.  

Answer:thanks for your comments. According to your suggestion, we have changed the reference in the manuscript.

[25] T.C. Coutinho, M.J. Rojas, P.W. Tardioli, E.C. Paris and C.S. Farinas, Nanoimmobilization of beta-glucosidase onto hydroxyapatite, International journal of biological macromolecules, 119 (2018), 1042-1051.

(5) In figure 4, the scale of the SEM images are different. In the general view, it seems that dextranase is fully immobilized, while HA is partially immobilized, however, its magnification is 100 times smaller. Please choose the same magnification and change it.

Answer:thanks for your comments. According to your suggestion, we have added same magnification SEM in the manuscript.

(6) In section 3.6, the authors mentioned that “Crystal violet staining in the present experiment revealed that immobilized dextranase was the best for removing dental plaques (Figure 6). Interestingly, HA was found to be more effective in dental plaque removal than dextranase.”. According to Table 1 and Figure 7, the dextranase immobilized sample was more effective among all, which is confirmed by the first sentence above. However, the second sentence refers to HA was found the most effective. Please correct this mismatch.

Answer:thanks for your comments. According to your suggestion, we have corrected in the manuscript.

(7) Table 1, Figure 3, Figure 5a and Figure 5b were not mentioned in the text. Please add relevant sentences to refer to them.

Answer:thanks for your comments. We have added Table 1, Figure 3, Figure 5a and Figure 5b with relevant sentences in the manuscript.

(8) The conclusion part is missing in the manuscript. Please add it.

Answer:thanks for your comments. We have added conclusion in the manuscript.

Reviewer 2 Report

In this manuscript authors have immobilized dextranase on hydroxyapatite (HA) nanoparticle and studied the conditions, properties, utilization, and storage stability of it. Overall, the manuscript is good but, in my opinion language of the manuscript would have been better, and authors should have shown more application of it along with biofilm inhibition. Have authors tried immobilization on other nano material as well?

Author Response

Dear Editor and Reviewers,

We have read these comments carefully, and we are really appreciated for you kindly comments. We have revised our manuscript according to the suggestions. The amend parts have been marked as yellow background in the manuscript. The main corrections in the paper and the responds to the reviewer’s comments are as following:

Have authors tried immobilization on other nano material as well?

Answer:thank you very much for your comments. Yes, we have immobilized dextranase on nano layer double hydroxides (LDH), and we focused on relationship between amino acid of dextranase to LDH in the study. In this manuscript, we tried to investigate the application.

Reviewer 3 Report

The aim of the manuscript “Immobilization of dextranase on nano-hydroxyapatite as a recyclable catalyst” was to assess the characteristics of dextranase immobilized onto hydroxyapatite nanoparticles for future use and for use in dental plaque biofilm removal.

The paper is well written and the research is interesting. However some drawbacks need to be addressed, as follows:

Materials and Methods

Please specify how dextranase was prepared in the laboratory. Also, please specify the other reagents not just “All other reagents were of analytical grade” (row 71-72).

Results

Figure 3 – FEG-SEM images a (FEG-SEM images of HA) and b (FEG-SEM images of immobilized dextranase) are obtained with different magnifications (500 and 5000 respectively) and cannot be compared. Please provide FEG-SEM images of HA with higher magnification.

Discussion paragraph is missing.

Also clearly stated conclusions are mandatory.

Author Response

Dear Editor and Reviewers,

We have read these comments carefully, and we are really appreciated for you kindly comments. We have revised our manuscript according to the suggestions. The amend parts have been marked as yellow background in the manuscript. The main corrections in the paper and the responds to the reviewer’s comments are as following:

1.Materials and Methods
Please specify how dextranase was prepared in the laboratory. Also, please specify the other reagents not just “All other reagents were of analytical grade” (row 71-72).

Answer:thanks for your comments. According to your suggestion, we have added relevant content in the manuscript.

2.Results
Figure 3 – FEG-SEM images a (FEG-SEM images of HA) and b (FEG-SEM images of immobilized dextranase) are obtained with different magnifications (500 and 5000 respectively) and cannot be compared. Please provide FEG-SEM images of HA with higher magnification.

Answer:thanks for your comments. According to your suggestion, we have detected the sample with SEM and added relevant content in the manuscript.

3.Discussion paragraph is missing.
Also clearly stated conclusions are mandatory.

Answer:thanks for your comments. According to your suggestion, we have added conclusions in the manuscript.

Reviewer 4 Report

Y. Ding and co-authors developed a trivial method to immobilize dextranase on nano-HA and tested its plaque degrading ability. Unfortunately, the manuscript has several shortcomings

  1. Language: editing required. The construction of some sentences is odd.
  2. Figure 1
    - fig. d) pH 4: low enzyme activity --> enzyme denaturated at pH4? What is the effect of pH 4 on nano-HA?
    - caption: a) add time, b) add temperature d) pH (not Ph)
  3. Desorption by sodium salts:
    - compare the pKa values of the relevant acids.
    - L182 - 191: The authors discuss the impact of ionic bonds. However, a lower pH improves immobilization, clarify. Furthermore the authors compare acetate and citrate. Counting the carboxylic groups is not convincing, consider the pKa values of the acids and the pI of dextranase. Furthermore, why not simply a calcium-citrate chelation effect?
    Fluoride (HF) is a very strong calcium binder --> CaF2
  4. Figure 4
    a-b) strong charging effect. A voltage of 15 is way too high. At 15 kV only the HA can be investigated. Moreover, both images cannot be compared due to different resolutions.
    c) no difference is visible?
    d) HA is the most stable phase at RT and somewhat neutral conditions -  and even if, a transformation at the surface can probably not be detected by XRD. Furthermore, the two XRD patterns should have a comparable intensity
  5. Table 1:
    - Control: If the average is 0.0, how can the stdev. be +/- 8.66 ?!
    - Explain precisely how the concentrations of the additives have been adjusted.
    - Tab. 1 and Fig. 6 overlap slightly

Author Response

Dear Editor and Reviewers,

We have read these comments carefully, and we are really appreciated for you kindly comments. We have revised our manuscript according to the suggestions. The amend parts have been marked as yellow background in the manuscript. The main corrections in the paper and the responds to the reviewer’s comments are as following:

Language: editing required. The construction of some sentences is odd.

Figure 1 
- fig. d) pH 4: low enzyme activity --> enzyme denaturated at pH4? What is the effect of pH 4 on nano-HA?
- caption: a) add time, b) add temperature d) pH (not Ph)

Answer:thank you very much for your comments. Our manuscript has been edited to the company which works on English edit.

The dextranase had showed lower activity at pH 4. It was not denaturated, and the pH 4 was not its optimal condition. So, the enzyme activity was lower. The nano-HA was not affected at pH 4. We mixed nano-HA with dextranase at pH 4, and the quantity of nano-HA was not decreased after centrifugation.  

  1. The reaction was taken 1 h.
  2. The reaction temperature was 25℃.

Desorption by sodium salts: 
- compare the pKa values of the relevant acids.
- L182 - 191: The authors discuss the impact of ionic bonds. However, a lower pH improves immobilization, clarify. Furthermore the authors compare acetate and citrate. Counting the carboxylic groups is not convincing, consider the pKa values of the acids and the pI of dextranase. Furthermore, why not simply a calcium-citrate chelation effect?
Fluoride (HF) is a very strong calcium binder --> CaF2

Answer:thanks for your comments. We have listed pKa of the acids in table 1. pKa of Fluoride and Citric acid is lower than Acetic acid, so they are strong competitor with Calcium of HA. Different forces are involved in adsorption of dextranase and HA, such as electrostatic forces between charged groups, van der Waals forces, and hydrogen bonds with OH− groups of hydroxyapatite, although it is difficult to determine their relative contributions. For the desorption, we think the main forces was ionic force.

Ivic et al. demonstrated the same interaction during immobilization of lipase on hydroxyapatite. One of the experiments performed to prove the interaction force.  The results showed that the coordination bonds between COO− and Ca2+ led to high binding affinity of the enzyme to the HA matrix[1].

[1] J. Ivic, A. Dimitrijevic, N. Milosavic, D. Bezbradica, B. Drakulic, M. Jankulovic, M. Pavlovic, H. Rogniaux, D. Velickovic, Assessment of the interacting mechanism between Candida rugosa lipases and hydroxyapatite and identifification of the hydroxyapatite-binding sequence through proteomics and molecular modelling, RSC Adv. 6 (41) (2016) 34818–34824.

Table 1 pKa of different acids

name

pKa      pKa2 

HCl

-8.00 

HF

3.18

HOOCOHC-(CH2COOH)2

3.13      4.76

CH3COOH

4.74

Figure 4 
a-b) strong charging effect. A voltage of 15 is way too high. At 15 kV only the HA can be investigated. Moreover, both images cannot be compared due to different resolutions.
c) no difference is visible?
d) HA is the most stable phase at RT and somewhat neutral conditions -  and even if, a transformation at the surface can probably not be detected by XRD. Furthermore, the two XRD patterns should have a comparable intensity

Answer:thanks for your comments. We have redone the SEM at 3 kV, and the pictures were in same resolutions.

Fig1 (a) FEG-SEM images of HA, (b) FEG-SEM images of immobilized dextranase

The main peaks of HA and HA-enzyme was same, however, they were slightly different shift. For example, the peak of 1631 and 1420 were in HA, but there were 1648 and 1421 in HA-enzyme.    

The XRD results showed the crystal of HA and HA-enzyme, and the peaks were same. We think the intensive expressed the crystal modal, and they did not have a significate differ.

Table 1: 
- Control: If the average is 0.0, how can the stdev. be +/- 8.66 ?! 
- Explain precisely how the concentrations of the additives have been adjusted.
- Tab. 1 and Fig. 6 overlap slightly

Answer:thanks for your comments. For the experiment of biofilm removal, three groups of parallel experiments were carried out to take the average value, and the error value calculated by the three results was 8.66. Then, we calculated the ratio of removal, and the ratio of removal of control was zero. 

According to our previous experiment, 10 mg HA could absorb 3.6 U dextranase. We added 10 mg HA-enzyme in 200 μL distilled water, similarly, we diluted and adjusted concentration of dextranase as 3.6 U per 200 μL.

We agree with you that Tab. 1 and Fig. 6 are overlap. According to your comments, we have moved Fig. 6 to supporting information.

Reviewer 5 Report

1. Figure 2a vs 2b and Figure 2c vs 2d - what is the difference? The labels are randomly switched… experiment is the same - activity as a function of pH and temperature, but the data is different? This data and conclusions are not clear at all.

2. This paper suffers from a lack of electrochemical data on this system. Enzymes act as bioelectrocatalysis when immobilized on electrode surfaces - so electrochemical data is key in terms of immobilization strategies for enzymes.

3. The applications for the presented immobilization are not clear, therefore it is difficult to understand the significance of this study.

4. S. mutans introduced only in results section in terms of dental plague. This section of the paper is somewhat out of place as it is not introduced - what is the significance? This needs to be introduced in the introduction.

5. The authors state that HA was found to be more effective in dental plague removal than dextranase? Why? But immobilized dextranase achieved biofilm degradation? Dental plaque is a biofilm of microorganisms. Also, the authors only test S. mutans while there are thousands of microorganisms in the oral microbiota. What about polymicrobial environments where biofilms are composed of multiple species which would closely represent real environments.

6. How do the authors know these are biofilms? SEMs are shown requiring to dehydrate cells, so imaging is likely of dead cells. Calculating the biofilm removal rate from SEM images is thereby tricky as most of the bacteria are likely dead at the time of imaging. Confocal microscopy imaging is necessary here to characterize biofilm and effects of HA and dextranase.

Author Response

Dear Editor and Reviewers,

We have read these comments carefully, and we are really appreciated for you kindly comments. We have revised our manuscript according to the suggestions. The amend parts have been marked as yellow background in the manuscript. The main corrections in the paper and the responds to the reviewer’s comments are as following:

  1. Figure 2a vs 2b and Figure 2c vs 2d - what is the difference? The labels are randomly switched… experiment is the same - activity as a function of pH and temperature, but the data is different? This data and conclusions are not clear at all.

Answer:thanks for your comments. We compared the characteristics of dextranase and immobilized one. Figure 2 was the results. (a) pH stability, (b) optimal pH of activity, (c) temperature stability, and (d) optimal temperature of activity.

  1. This paper suffers from a lack of electrochemical data on this system. Enzymes act as bioelectrocatalysis when immobilized on electrode surfaces - so electrochemical data is key in terms of immobilization strategies for enzymes.

Answer:thanks for your comments. According to your suggestion, we have added data of ζ-potential of HA in the manuscript.

HA nanoparticles show a negative potential in the pH range of 4-9. The PI of dextranase is 5.19. So, when the processing was performed in acidic condition (pH 4, and pH 5), the ratio of immobilization was higher. The fixation effect was the best when the pH is 5.  

  1. The applications for the presented immobilization are not clear, therefore it is difficult to understand the significance of this study.

Answer:thanks for your comments. The dextranase has been applied prospects in diverse industries, such as in the production of low molecular-weight medical dextran, food processing, sugarcane factories, and prevention of dental plaque biofilm. Also, Hydroxyapatite is a non-toxic nano-material, and it is a promising inorganic material. The immobilized dextranase will have a broad application in food, medicine, chemical industries.

  1. S. mutansintroduced only in results section in terms of dental plague. This section of the paper is somewhat out of place as it is not introduced - what is the significance? This needs to be introduced in the introduction.

Answer:thanks for your comments. According to your suggestion, we have added relevant content in the introduction.

  1. The authors state that HA was found to be more effective in dental plague removal than dextranase? Why? But immobilized dextranase achieved biofilm degradation? Dental plaque is a biofilm of microorganisms. Also, the authors only test S. mutanswhile there are thousands of microorganisms in the oral microbiota. What about polymicrobial environments where biofilms are composed of multiple species which would closely represent real environments.

Answer:thanks for your comments. Dental plaque is a biofilm structure composed of a variety of microorganisms, the internal microbial community is mainly composed of streptococcus, actinomyces, Lactobacillus and other microbial cells. In the early stage of the formation of dental plaque biofilm, streptococcus is the main bacterium[1]. S. mutans could secret dextransucrase which could produce dextran by using sucrose in oral. S. Mutans is often used as a model strain for plaque biofilm construction and study. The dextran, stuck on surface of teeth, is the basic structure of dental plaque biofilm. Therefore, the dental plaque will be removed if the structure of dextran was degraded.

[1]Al-Ahmad A, Wunder A, Mathias Auschill1 T, Follo M, Braun G, Hellwig E, Arweiler N. The in vivo dynamics of Streptococcus spp., Actinomyces naeslundii, Fusobacterium nucleatum and Veillonella spp. in dental plaque biofilm as analysed by five-colour multiplex fluorescence in situ hybridization. Journal of medical microbiology, 2007,56(5):681-687

  1. How do the authors know these are biofilms? SEMs are shown requiring to dehydrate cells, so imaging is likely of dead cells. Calculating the biofilm removal rate from SEM images is thereby tricky as most of the bacteria are likely dead at the time of imaging. Confocal microscopy imaging is necessary here to characterize biofilm and effects of HA and dextranase.

Answer:thanks for your comments. We cultured S. mutans in 24 wells plates, and a glass was added in each well. We washed and detected the surface of glass by SEM. The biofilms stuck on the surface and could not be washed away. When we calculated the biofilm removal rate we used 96 wells plate. The crystal violet stain method was used.

Yes, confocal microscopy imaging could express the live and dead cells. In our experiments, we focused the degradation of dextran. The bacteria cannot stuck on the surface glass without the biofilm constructed by dextran.

Round 2

Reviewer 3 Report

The manuscript was improved and the authors partially addressed the reviewers' comments.

However, further discussions are required.

Author Response

Dear Editor and Reviewers,

We have read the comments carefully, and we are really appreciated for you kindly comments. We have revised our manuscript according to the suggestions. The amend parts have been marked as yellow background in the manuscript. The main corrections in the paper and the responds to the reviewer’s comments are as following:

The manuscript was improved and the authors partially addressed the reviewers' comments.

However, further discussions are required.

Answer:thanks for your comments. According to your suggestion, we have discussed the application of immobilized enzyme.

Sincerely Yours,

Yanshuai Ding

Jiangsu Key Laboratory of Marine Bioresources and Environment /Jiangsu Key Laboratory of Marine Biotechnology, Jiangsu Ocean University

59 Cangwu Road, Lianyungang, 222005, China.

Tel: +13963700556;
Fax: +8651885891573;

Reviewer 4 Report

" Our manuscript has been edited to the company which works on English edit": I still think that language editing is required.

L70: ...were investigated. and....
L123, methods: explain the preparation procedure for SEM
L176, wording: The highest 176 immobilized yield was performed at pH 5
L178, wording: ...the dextranase was weak negative charge at pH 5.
L225, wording: The surface of HA showed smooth after immobilization of dextranase
Fig. 4 a,b: The SEM image quality improved but no real difference can be detected. Furthermore, the authors state that immobilization of dextranase increases the tendency of particle agglomeration. Correct? But also the nano-HA control is clotted. I assume that the SEM specimen were obtained by centrifugation which is obviously the wrong approach.
L240: The authors claim that immobilization did not alter the crystal form (phase?). However, the XRD patterns have a low intensity (noisy) and are not indexed. A differentiation with other putative mineral phases like OCP is barely feasible. For instance, the distinctive presence of the (100) line of OCP at very low 2θ angles (<10°) would be helpful for distinguishing OCP from HA. To this end, the diffractometer setup must sometimes be adapted.
Table 1:
- Why does HA exert such a strong anti-biofilm effect?
- The data shows that the removal rate of immobilized dextranase is not an additive effect (free enzyme plus HA). Is it a synergistic effect? Explain.
- How can a simple water control have an error of +/- 8.6 while the rather complex series of experiment with immobilized dextranase has an error of about +/- 1?

Supporting information: Zeta, not Zate

Author Response

Dear Editor and Reviewers,

We have read the comments carefully, and we are really appreciated for you kindly comments. We have revised our manuscript according to the suggestions. The amend parts have been marked as yellow background in the manuscript. The main corrections in the paper and the responds to the reviewer’s comments are as following:

" Our manuscript has been edited to the company which works on English edit": I still think that language editing is required.

L70: ...were investigated. and....

L123, methods: explain the preparation procedure for SEM

L176, wording: The highest 176 immobilized yield was performed at pH 5

L178, wording: ...the dextranase was weak negative charge at pH 5.

L225, wording: The surface of HA showed smooth after immobilization of dextranase
Answer:thanks for your comments. According to your suggestion, we have added the preparation procedure for SEM in the manuscript. The whole manuscript has been edited before the first submit. However, we did not send the revised manuscript to reedit. We have sent the revised manuscript for reediting.

Fig. 4 a,b: The SEM image quality improved but no real difference can be detected. Furthermore, the authors state that immobilization of dextranase increases the tendency of particle agglomeration. Correct? But also the nano-HA control is clotted. I assume that the SEM specimen were obtained by centrifugation which is obviously the wrong approach. 

Answer:thank you very much for your comments. We have re-done the SEM. The solution of HA and HA + dextranase was dropped on glass slides and lyophilized. The results of SEM showed that immobilization of dextranase increased the tendency of particle agglomeration when the solution was dried.

(a)SEM of HA, (b) SEM of HA + dextranase

L240: The authors claim that immobilization did not alter the crystal form (phase?). However, the XRD patterns have a low intensity (noisy) and are not indexed. A differentiation with other putative mineral phases like OCP is barely feasible. For instance, the distinctive presence of the (100) line of OCP at very low 2θ angles (<10°) would be helpful for distinguishing OCP from HA. To this end, the diffractometer setup must sometimes be adapted.
Answer:thanks for your comments. We performed the XRD experiment again and got the latest data. After immobilization, the crystal structure of HA has not changed [1-3]. The diffraction pattern shows that the intensity of HA immobilized enzyme is lower than that of HA. Thank you very much for your suggestion.

 The XRD patterns of  HAand HA-Enzyme

[1]T.C. Coutinho, P.W. Tardioli, C.S. Farinas, Hydroxyapatite nanoparticles modifified with metal ions for xylanase immobilization, Int. J. Biol. Macromol. 150 (2020) 344–353.

[2] M. Moran-Pineda, S. Castillo, T. Lopez, R. Gomez, O. Novaro, Synthesis, characterization and catalytic activity in the reduction of NO by CO on alumina–ZrO2 sol–gel derived mixed oxides, Appl. Catal. B Environ. 21 (1999) 79–88.

[3] S.F. Mansour, S.I. El-Dek, M.K. Ahmed, Physico-mechanical and morphological features of ZrO2 substituted hydroxyapatite nano crystals, Sci. Rep. 7 (2017), 43202, .

Table 1: 
- Why does HA exert such a strong anti-biofilm effect? 
- The data shows that the removal rate of immobilized dextranase is not an additive effect (free enzyme plus HA). Is it a synergistic effect? Explain.
- How can a simple water control have an error of +/- 8.6 while the rather complex series of experiment with immobilized dextranase has an error of about +/- 1?

Answer:thanks for your comments. HA is a great abrasive, and it is added in toothpaste. The experiments were performed on a shaker, so, the HA showed a strong anti-biofilm effect. We think the immobilized dextranase has synergistic effect because the biofilm (dextran) was hydrolyzed by dextranase and easier to be removal. For the error bar of control, we used water in the control. And results of experiments were detected after incubation for 24 h. The error bar was calculated according to the results. We thought that the culture of bacteria was the main reason for the different error bar. Biological samples might cause error bar compare to chemical samples. 

Supporting information: Zeta, not Zate

Answer:thank you very much for your comments. We have revised it in the supporting information. 

Sincerely Yours,

Yanshuai Ding

Jiangsu Key Laboratory of Marine Bioresources and Environment /Jiangsu Key Laboratory of Marine Biotechnology, Jiangsu Ocean University

59 Cangwu Road, Lianyungang, 222005, China.

Tel: +13963700556;
Fax: +8651885891573;

Reviewer 5 Report

The authors have addressed the questions, therefore, this reviewer recommends the manuscript for publication.

Author Response

Dear Editor and Reviewers,

We have read the comments carefully, and we are really appreciated for you kindly comments. We have revised our manuscript according to the suggestions. The amend parts have been marked as yellow background in the manuscript. The main corrections in the paper and the responds to the reviewer’s comments are as following:

The authors have addressed the questions, therefore, this reviewer recommends the manuscript for publication.

Answer:thank you very much for your comments.

Sincerely Yours,

Yanshuai Ding

Jiangsu Key Laboratory of Marine Bioresources and Environment /Jiangsu Key Laboratory of Marine Biotechnology, Jiangsu Ocean University

59 Cangwu Road, Lianyungang, 222005, China.

Tel: +13963700556;
Fax: +8651885891573;
